# Where does the carbon go? A new carbon balance method to assess what happens to plastics under solar exposure

Gustave Bertier[1,2], Arnaud Martel[2,3], Matthieu George[1], Pascale Fabre[1],
Fabien Boucher[3], Justine Gérome[3], Fabienne Lagarde[2,3]*

**1** Laboratoire Charles Coulomb (L2C), Université de Montpellier, CNRS, Montpellier, France, **2** Institut des Molécules et Matériaux du Mans - Université du Mans, Le Mans, France, **3** IUT du Mans - Université du Mans, Le Mans, France

* Fabienne.Lagarde@univ-lemans.fr

## Abstract

Plastic pollution is a major and global threat to ecosystems and human health, resulting from the spreading and breakdown of plastic litter in the environment. In an aquatic environment, the first causes of this degradation are exposure to natural ultraviolet light and abrasion or collisions in the water. The extent of such degradation on a plastic object after a given time remains very difficult to quantify, especially regarding the relative production of microplastics, nanoplastics and soluble species along with volatile compounds. All of these degradation products may contribute differently to environmental pollution. Therefore, when evaluating the pollution caused by plastic objects, we should consider how much of each byproduct is generated. We propose a novel method based on conservation of the carbon mass during the degradation process. This approach is the first to enable quantification of carbon retrieved in each type of degradation product (Microplastics, Nanoplastics, Solubles, Volatile Compounds), as well as its evolution with exposure time. By applying this method to polypropylene granules, we demonstrate its effectiveness in tracking carbon footprint throughout the aging process. One of the unexpected results of this study is to show that the amount of carbon released in volatile form is far from negligible (17%) compared to MP (55%). The procedure we present is general enough to be applied to any type of polymer, and can be a valuable tool for assessing the amount of by-products of a given size released into the environment.

## Introduction

Since their initial use in the 1950s, plastics have accumulated in the environment to such an extent that already in 1997, during a study on marine debris, a vast quantity of plastic items commonly appeared [1,2]. Since then, all over the world, numerous

**Data availability statement:** All relevant data are within the manuscript and its Supporting Information files.

**Funding:** PNR-EST CIPROP funded by ANSES.

**Competing interests:** The authors have declared that no competing interests exist.

studies have confirmed that plastic debris are present, even in the most remote areas of the planet [3–5].

Plastics are composed of different types of polymers, which are combined with various chemical components, known as additives, to create unique materials with specific properties. Consequently, a multitude of plastic types can be produced through a variety of combinations, resulting in noticeably varied environmental degradation behaviors [6]. Moreover, depending on the environment in which plastics accumulate and degrade over time, very different degradation processes will take place [7,8]. Despite their relative stability, plastics are susceptible to damage from long-term exposure to UV light, which is a primary driver of abiotic plastic degradation in the environment. Photo-degradation of plastic debris leads to structural and property changes, ultimately producing a wide range of degradation products. When photo-degradation is coupled with mechanical abrasion, breakdown or surface delamination processes result in the release of smaller plastic particles [9–11], commonly defined as microplastics (size $\in$ [1µm; 5 mm] [12]) and nanoplastics (size $\in$ [1nm; 1µm] [13]). Microplastics (MP) and nanoplastics (NP) have been extensively studied and their presence in all environmental and biological compartments is now commonly recognized [14–16]. More recently, the release of volatile organic compounds (VOCs) resulting from chemical degradation, induced in particular by heat and UV-light, has begun to draw the attention of scientists [17,18]. As plastics degrade, the polymer chain shortens, loses weight, changes its molecular composition, and emits greenhouse gases such as $CO_2$ (carbon dioxide), $CH_4$ (methane) or other volatile molecules [17,19]. The process of VOC release associated with photo-degradation of polymers is long-standing and well known [20]. However, given the growing amount of plastic debris stranded or stored worldwide in the environment and exposed to solar heat and daylight, the emission of such VOCs may now represent a considerable threat for the environment, climate and human health [21–23]. Additionally, under photo-degradation, the dissociation of polymer chains into low-molecular weight, water-soluble organic molecules is a further possible outcome, which has already been measured in different types of plastics [18,24]. The production of small organic acids and solubilizable $CO_2$ gives rise to an additional concern regarding ocean acidification [25], one of the planetary boundary threats in which the crossing of certain critical thresholds could be very harmful to societies throughout the world and, more particularly, for the integrity of the global ecosystem. Microplastics and organic pollutants are also partially responsible for the crossing the planetary boundaries with regard to novel entities [26,27].

Depending on pristine plastic, the path and point of arrival of plastic particles is strongly linked to their size [28]. The products resulting from plastic degradation possess a range of physico-chemical properties and composition characteristics affording them the potential to be transported by several means, including water or air currents and animal transportation. It therefore seems to be of utmost importance to assess and quantify the many different types of products generated by plastic degradation in the environment, the objective being to fully predict their impacts. Even when the specific characteristics of these products are understood, their diversity

makes it very difficult to monitor all of them, especially insofar as distinct analytical protocols adapted to the characterization and quantification of particles, gas or soluble compounds are required.

When conducted under UV light, the photo-oxidation process results in the formation of hydroxyl, ketone, or carboxylic acid groups. This modification leads to a change in the molar mass of the polymer, which is marked by the introduction of oxygen atoms [29–31] and an increase during aging of the total mass (or amount of substance) of plastic derivatives. It is therefore not feasible to conduct a mass balance of all degradation products. In contrast, since the carbon mass present in the original pristine plastic being studied is equal to the sum of the carbon mass in all abiotic degradation products, an assessment of carbon mass evolution is the most reliable method for evaluating the complete degradation of plastics, regardless of their chemical structure.

Notwithstanding the existence of numerous studies examining plastics' degradation products, to the best of our knowledge no study has proposed a procedure to achieve a comprehensive assessment of all the products released during the abiotic degradation of plastics. This article presents the fundamental principles of a carbon balance assessment of abiotic degradation and provides a detailed account of the methodologies employed to conduct such assessment. In addition, the results obtained on aged polypropylene (PP) industrial granules are presented. Mainly used for short- term applications such as packaging, PP is one of the most abundant industrially produced plastics, representing around 19% of the rapidly growing quantity of plastics produced worldwide [32–35]. Mostly by shipping, industrial granules travel all around the world, and some of them regularly wash up on beaches and other shores, where they may stay for decades without their ultimate fate being fully known [36,37].

In this study, the term "aging" is used to describe the effects of ultraviolet (UV) radiation, while "weathering" encompasses both aging and the effects of stirring in water. Pristine plastic granules (G) were placed in quartz containers and subjected to UV light for specified times, during which they underwent an aging process. Subsequently, the UV-aged plastic granules (GUV) were immersed in ultrapure water and subjected to mechanical stirring with a magnetic rod in order to simulate mechanical abrasion. This weathering protocol was selected to emulate the primary factors that contribute to the degradation and transformation of plastics in the environment. Following sieving and filtration, the remaining quantities of carbon in the granules and those released in the form of microplastics, nanoplastics and water-soluble compounds, together with volatile products leaking in the atmosphere during photo-degradation, were evaluated over several durations of aging. To this end, a series of procedures was undertaken, including mass monitoring, elemental analysis, total organic carbon (TOC) measurements and, for each UV dose, a comprehensive carbon balance assessment of the degradation process,. The resulting findings offer preliminary insights into the long-term fate of PP granules when discarded in the environment over an extended time period.

## Experimental section

The primary objective of this study is to develop and optimize a protocol for assessing the carbon balance of PP granules subjected to weathering. This shall be achieved by quantifying the masses and carbon contents (or carbon fractions) of the granules and their by-products. While the total mass of plastic is not conserved during degradation due to oxidation and modification of the carbon backbone, the mass of carbon present in the initial pristine plastic granules $(mC_{G,t0})$ is conserved at any time t, and is therefore equal to the sum of the carbon mass in each degradation product $(mC_{Total,t})$.

The degradation by-products are sorted into four categories that the designed protocol separately quantifies;

- volatile compounds (Vol,t) released during UV exposure

- nanoplastic/soluble compounds (NP/Sol,t), which include all the compounds released into the water during stirring and with size below 1 μm. They are at once soluble molecules and small fragments.

- microplastics (MP,t) generated during stirring and with size exceeding 1 μm

- remnants of plastic granules after the whole weathering process: UV exposure and stirring in water (GUVω,t),

## Samples

The plastics used in this experiment are industrial granules of polypropylene sourced from Prospector® (Sabic® PP 531Ph). They exhibit a density of 0.905g.cm-3 and the average size of the granules is x=4.64±0.23mm, y=4.61±0.17mm and z=3.25±0.24mm. The medical grade of the polymer accounts for a very small amount of additives (<1w%).

## Overall description of the protocol

The sequence of the protocol is described in this subsection and summarized in Fig 1. Specific details on each experimental technique are given in the subsequent subsections. A comprehensive list of the abbreviations used is given in Table 1.

The samples, of approximately two grams (or about 70 granules) of extruded pristine polypropylene granules are disposed in one layer in cylindrical quartz containers covered with non-air-tight lids to limit sample pollution while not inhibiting $O_2$ access for the plastic. These containers had previously been through a three-stage cleaning process, including ethanol washing, Milli-Q ultrapure water rinsing, and drying. The sample are weighted ($m_{G, t0}$) then placed in an artificial UV chamber. The samples are removed from the chamber at different aging times (15, 23, 27, 44, and 52 days) and once again weighted ($m_{GUV,t}$). During aging, the carbon mass released in the form of volatile compounds will be noted $mC_{Vol,t}$.

For each UV exposure time, 8 granules were extracted from the sample, weighted and stirred in water with a magnetic stirrer at a nominal rotation speed of 990rpm. Each magnetic stirrer was thoroughly washed before use and no sign of wear or degradation was observed. The goal of the stirring is to create a vortex that brings the plastic granules to the bottom of the vial, in contact with the magnetic rod, causing mechanical abrasion and erosion of the polymer granules by mutual collision in the vial. This second step of the protocol generates the three other categories of degradation products considered in this study (Fig 1), representing an approach to the action of wave and abrasion in the environment:

- the remnants of the base plastic granules after weathering operations (noted GUVω,t). While they could technically be considered microplastics (size<5mm) [12], they are studied separately here.

- the small microplastics (noted MP,t)

- the nanoplastics and soluble compounds (NP/Sol,t).

The remnants of the eight polypropylene (PP) granules were removed from the solution using tweezers and transferred to a glass container of known mass. This container with the plastics remnants was dried in an oven at 60°C for 24 hours and, subsequently, weighted to determine the mass of the degraded granules ($m_{GUVω,t}$).

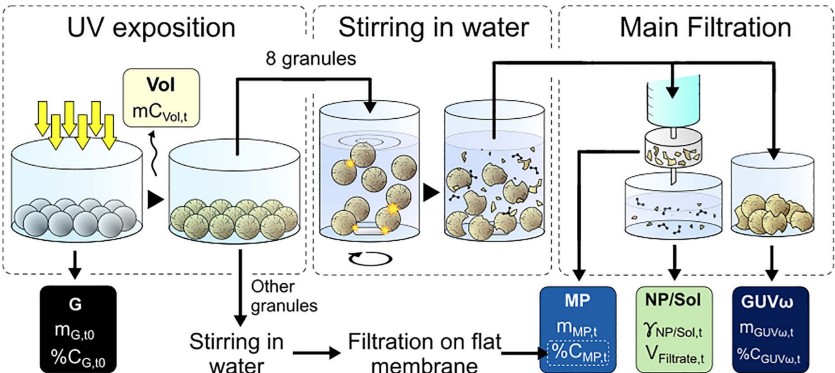

**Fig 1. Description of the complete protocol for carbon balance assessment.**

**Table 1. Name, symbol and dimension of the variables used.**

| When | Name | Symbol | Dimension (Unit) | Acquisition |
|---|---|---|---|---|
| Initial state | Carbon mass in Initial Granules | $mC_{G,t0}$ | Mass (mg) | Calculation |
| | Mass of Initial Granules | $m_{G,t0}$ | Mass (mg) | Weighting |
| | Carbon fraction of Plastic Granules | $\%C_{G,t0}$ | Dimensionless | Elemental analysis CHNS |
| After UV exposure | Carbon mass in Granules after UV Exposure | $mC_{GUV,t}$ | Mass (mg) | Calculation |
| | Mass of Granules after UV Exposure | $m_{GUV,t}$ | Mass (mg) | Weighting |
| | Carbon mass fraction of Granules after UV Exposure | $\%C_{GUV,t}$ | Mass (mg) | Elemental analysis CHNS |
| After stirring in water | Carbon mass in all of the Degradation Products | $mC_{Total,\ t}$ | Mass (mg) | Calculation |
| | Proportion of initial Carbon retrieved in Remnants of Plastic Granules after UV Exposure and stirring | $pC_{GUV\omega,t}$ | Dimensionless | Calculation |
| | Carbon mass in Remnants of Granules after UV Exposure and Stirring | $mC_{GUV\omega,t}$ | Mass (mg) | Calculation |
| | Mass of Remnants of Granules after UV Exposure and Stirring | $m_{GUV\omega,t}$ | Mass (mg) | Weighting |
| | Carbon fraction of Remnants of Granules after UV Exposure and Stirring | $\%C_{GUV\omega,t}$ | Dimensionless | Elemental analysis CHNS |
| | Proportion of initial Carbon retrieved in Microplastics | $pC_{MP,t}$ | Dimensionless | Calculation |
| | Carbon mass in Microplastics | $mC_{MP,t}$ | Mass (mg) | Calculation |
| | Mass of Microplastics | $m_{MP,t}$ | Mass (mg) | Weighting |
| | Carbon fraction of Microplastics | $\%C_{MP,t}$ | Dimensionless | Elemental analysis CHNS |
| | Proportion of initial Carbon retrieved in Nanoplastics and Soluble Compounds | $pC_{NP/Sol,t}$ | Dimensionless | Calculation |
| | Carbon mass of Nanoplastics and Soluble Compounds | $mC_{NP/Sol,t}$ | Mass (mg) | Calculation |
| | Carbon mass Concentration of Nanoplastics and Soluble Compounds | $\gamma_{NP/Sol.,t}$ | Mass concentration (mg/L) | Total Organic Carbon |
| | Volume of the Filtrate | $V_{Filtrate}$ | Volume (mL) | Weighting |
| Deduced afterward | Proportion of initial Carbon in the form of Volatile Compounds | $pC_{Vol,\ t}$ | Dimensionless | Calculation |
| | Carbon mass in Volatile Compounds | $mC_{Vol,t}$ | Mass (mg) | Calculation |

The stirred solution is filtered by means of a 1μm pore size syringe filter to recover the MP. This filter is then dried and weighted. Its mass increase gives the MP mass generated by the granules ($m_{MP,t}$) during weathering. To account for the mass lost by the filter itself during the filtration process, we utilized blanks using the same amount of ultrapure water, and used the mass difference found (0.641 mg) to correct $m_{MP}$ (see supplementary information S1 Fig 2 in S1 SupInfo).

The last category of degradation products, i.e., nanoplastics and soluble compounds, is what remains in the solution that went through the filter. This filtrate volume ($V_{Filtrate,t}$) was measured and its carbon mass concentration ($\gamma_{NP/Sol.,t}$) was determined by TOC analysis. We used a blank sample containing only ultrapure water and undergoing the same exact protocol to estimate any possible carbon pollution and correct the values of $\gamma_{NP/Sol.,t}$ obtained. We used elemental analysis to find the carbon content in the initial granules and remnants of plastic granules ($\%C_{G,t0}, \%C_{GUV\omega,t}$) on the one hand and of MP ($\%C_{MP,t}$) on the other hand.

## Accelerated ageing chamber

UV exposure is performed with a SUPRATEC ultraviolet high-pressure lamp HTC 400−221 (400 W, 50 Hz) with a spectral range of 250–450 nm (for characteristics of the lamp, see supplementary information S1 Fig 3 in S1 SupInfo). The

 

temperature varies during the exposure from 18°C at the beginning to 63°C after a long exposure time. Due to the high energy of the UV lamp used, the samples were not placed directly underneath. Therefore, to counterbalance the lack of standardization of the UV exposure simulation chamber, the actual effect of aging under the UV lamp was tracked using the carbonyl index of the UV aged plastic granules (GUV) rather than the aging time itself.

## Stirring

Stirring is performed for 24h in a vial (WICOM 40 ml, EPA, transparent glass, 27.5 × 95 mm) filled with 35 mL of Milli-Q water with a magnetic stirrer (DLAB MS-M-S10). Each magnetic stirrer was thoroughly washed before use and no sign of degradation, or any wear was observed after use. The magnetic stirrers being nevertheless covered in Polytetrafluoro-ethylene (PTFE), the TOC results obtained on blanks (just water) that underwent the same protocol as the studied sample (ageing + stirring) were subtracted to avoid counting potential contamination during the NP assessment. It should be noticed that this correction quickly represents less than 1% of the amount of carbon retrieved in NPs after PP ageing.

## Filtration

Filtration is done using a 20 mL Luer-Lock Henke-Ject® plastic syringe with a Sterican® 21Gx4¾" 0.8x120 mm needle and a Whatman GD/X 13 syringe filter: GF/B membrane, pore size 1 µm, diam. 13 mm. After filtration, the syringe, the needle and the glassware of the stirring step were placed in an oven at 60°C for 24h and weighted.

At the end of the filtration process, the filtrates are weighed to ascertain the mass of water remaining and to incorporate this value into the final carbon balance following TOC measurements.

## Weighting

Weighing was performed with a Mettler XS105 DualRange balance, with $2 \times 10^{-5}$ g precision for the weighting of pristine plastic granules, remnants of UV aged and stirred plastic granules, microplastics, and filtrate. We used a microscale Mettler WXTS3DU, with $10^{-6}$ g precision to weight the elemental analysis samples.

## Elemental analysis

A FlashSmart (Thermo Fisher Scientific) is used to determine the carbon contents in the sample. Data were processed using EagerSmart Ver 1.00 software. For CHNS analysis, a tin container is used. Sample sizes between 1.5 and 2.0 mg were weighted. The instrument was calibrated with certified 2,5-Bis(5-tert-butyl-2-benzoxazolyl)thiophene (BBOT) and certified sulphanilamide to which a few milligrams of $V_2O_5$ were added for sulphur analysis.

Since the sample mass for elemental analysis is limited to about 2 mg, the elemental analyses on pristine and weathered granules are performed after a cryo-milling step, using a cryogenic Freezer/Mill (Spex certiprep 6770). The obtained powder is dried at 60°C prior to elemental analysis to remove traces of water. Cryo-milling is essential to prevent any influence on the results of heterogeneous aging inside the granules.

## TOC analysis

Total Organic Carbon (TOC) analysis is made using a Multi N/C 3100 from Analytik Jena. Total Organic Carbon (TOC) was determined as the difference between Total Carbon (TC) and Inorganic Carbon (IC), TOC = TC – IC. All data are reported using MultiWin software Version 4.12.01.0000. The vials used as containers for the TOC analysis were previously put in an oven at 500°C overnight to remove any residual organic matter from the glassware. The method parameters were as follows: oven temperature: 750°C, number of measurements: 5–8 to ensure deviation from the mean lower than 2%, sample volume: 200 µL, rinse volume: 2 mL, Integration time: 300 s, stirring: 6.

Calibration is performed at variable concentrations and fixed volume with potassium hydrogen phthalate for total carbon calibration (TC) and with sodium hydrogen carbonate for inorganic carbon (IC). The calibrated IC range is 0.1 to 10 mg/L

in carbon, while the TC range is 1–800 mg/L in carbon in three ranges (see supplementary information S1 Fig 4 and 5 in S1 SupInfo). The calibrations were measured using appropriate TC and IC methods and implemented in the final TOC analysis method. TOC is measured as TOC = TC-IC. We used 3 mg/L as the uncertainty.

**Fourier-transform infrared spectroscopy (FTIR)**

Infrared analysis and the Carbonyl Index (CI) have been described as efficient tools for assessing surface oxidation [38] in polymers. The carbonyl index is defined as the area of the carbonyl band (~1750 cm$^{-1}$) divided by the area under the CH aliphatic bending band (~1400 cm$^{-1}$). Initially, in polyolefins such as PP, there is no oxygen and therefore no carbonyl group. As aging progresses, the carbonyl group amount increases, as does the CI [39]. In this study, this indicator is used to compensate for the inhomogeneity of UV aging in the UV chamber by providing an aging reference. Here, an ATR-FTIR Vertex 70V from Bruker is used to measure the surface carbonyl index of the samples after UV aging.

## Results and analysis

The conservation of carbon mass can be expressed as:

$$mC_{G,t0} = mC_{Total,t} = mC_{GUV\omega,t} + mC_{MP,t} + mC_{NP/Sol,t} + mC_{Vol,t} \tag{1}$$

The mass of carbon ($mC$) in each of these products is calculated after their physical separation, combining mass loss and analytical measurements. Mass measurements ($m$) and carbon content or fraction ($\%C$), determined by elemental analysis, give together the mass of carbon ($mC$) in GUVω, MP and volatile compounds. The mass of carbon in NP and soluble compounds is obtained by direct TOC measurement of carbon mass concentration $\gamma_{NP/Sol.,t}$ in the filtrate of volume $V_{Filtrate,t}$.

Eq. 1 can be expressed as a function of the measured quantities:

$$m_{G,t0} \times \%C_{G,t0} = m_{GUV\omega,t} \times \%C_{GUV\omega,t} + m_{MP,t} \times \%C_{MP,t} + \gamma_{NP/Sol,t} \times V_{Filtrate} + mC_{Vol,t} \tag{2}$$

In order to normalize the results, the data is represented for each degradation product, in terms of part of initial carbon (pC) (Eq. 3), defined as the ratio of the mass of carbon in the considered product divided by the mass of carbon initially present in the pristine granules.

$$pC_{GUV\omega,t} = \frac{mC_{GUV\omega,t}}{mC_{G,t0}}; \ pC_{MP,t} = \frac{mC_{MP,t}}{mC_{G,t0}}; \ pC_{NP/Sol,t} = \frac{mC_{NP/Sol,t}}{mC_{G,t0}}; \ pC_{Vol,t} = \frac{mC_{Vol,t}}{mC_{G,t0}} \tag{3 - a, b, c, d}$$

The data are expressed in this form to facilitate assessment of the overall carbon balance and to enhance comprehension of the processes by which carbon is distributed from the initial granules into the volatile, soluble, or solid parts at the conclusion of the weathering period. Ultimately, given the inherent heterogeneity of the UV chamber, we chose to present the findings as a function of the carbonyl index (CI) measured at the surface of the degraded pellets rather than as a function of the exposure time (see sections "Accelerated ageing chamber" and "Discussion").

**Evolution of carbon mass fraction in plastic granules (GUVω) and microplastics (MP) upon weathering**

Measurements of the carbon mass fractions in the exposed and stirred granules and the generated MP (respectively $\%C_{GUV\omega,t}$ and $\%C_{MP,t}$) are displayed in Fig 2 as a function of the oxidation level, quantified by the carbonyl index.

For pristine granules, the carbonyl index (CI) was found to be below the detection limit in FTIR, which confirms that the granules were not photo-oxidized before the study.

As photo-degradation takes place, the CI measured at the surface of the granules rises until reaching values close to 5, confirming the building of an oxidation layer. As oxidation increases, the carbon fraction appears to be constant for the first

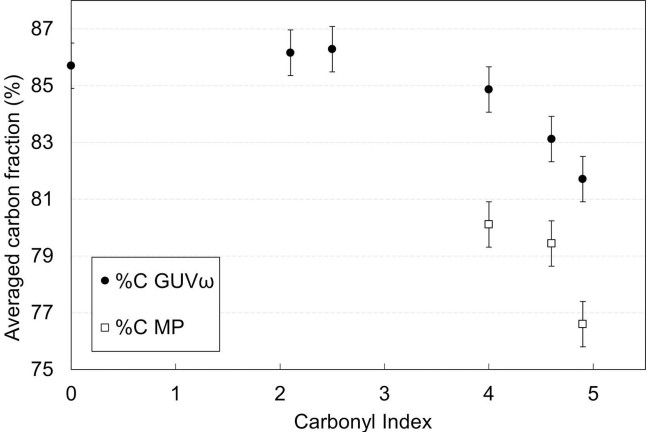

**Fig 2. Evolution of carbon mass fraction in remnants of granules and in microplastics as a function of post-aging carbonyl index.** The carbon fractions presented are normalized using the stoichiometric proportion of carbon in PP.

three measurement points, showing that the oxidation layer is not thick enough to affect the average carbon content of the entire granule. For CI ≥ 4, there is a clear overall decrease of the carbon fraction, confirming substantial thickening of the oxidized layer. MPs are obtained in sufficient amount to perform elemental analysis only for at least CI = 4. Their carbon fraction is lower than in the granule at the same aging time. This outcome aligns with the established understanding that MP are generated through the fragmentation of the outermost oxidized portion of the granules. The on-going decrease of the carbon fraction in MP nevertheless indicates that when delamination from the surface occurs, the MPs produced do not reach oxygen saturation.

Although this study focused on carbon content (expressed as a percentage, %C), elemental analysis also gave data on the nitrogen, sulphur, and hydrogen contents. The hydrogen content was seen to decline in a manner consistent with carbon content (see supplementary information S1 Fig 1 in S1 SupInfo), thereby substantiating the incorporation of oxygen into the material.

## Proportion of initial carbon in the form of volatile compounds

To allow enough oxygen to reach the plastic, UV exposure was carried out in dry quartz vessels with non-airtight lids. This also allowed volatile compounds to be released. Therefore, only the formation of volatile compounds can explain the loss of mass in the granules. More precisely, the difference in carbon mass of the PP samples before and after UV exposure indirectly measures the formation of volatile compounds. Indeed, a reasonable assumption is that a negligible quantity of volatile compound is produced during the stirring step, during which any molecular species generated would more likely, as will be confirmed later, be solubilized and accounted for as NP/Sol products. The mass of carbon transferred into volatile compounds can be expressed as:

$$mC_{Vol,t} = mC_{G,t0} - mC_{GUV,t} = m_{G,t0} \times \%C_{G,t0} - m_{GUV,t} \times \%C_{GUV,t} \tag{4}$$

It is not possible to measure the carbon fraction of the granules after UV exposure and before stirring $\%C_{GUV,\,t}$ without disturbing the carbon mass balance protocol. We nonetheless know that the carbon mass lost in NP/Sol is relatively small compared to MP or GUVω. Accordingly, we assume that the carbon mass remaining in the photodegraded granules ($mC_{GUV,t}$) can be determined by calculating the arithmetical mean between the carbon mass fractions of MP and the granule remnants, with their respective masses serving as the weights (Eq. 5, Fig 3).

$$mC_{GUV,t} = m_{GUV,t} \times \%C_{GUV,t} \approx m_{GUV,t} \times (\%C_{GUV\omega,t} \times \frac{m_{GUV\omega,t}}{m_{MP,t} + m_{GUV\omega,t}} + \%C_{MP,t} \times \frac{m_{MP,t}}{m_{MP,t} + m_{GUV\omega,t}}) \tag{5}$$

These results suggest that the release of carbon in volatile form occurs as early as a CI of 2.5, reaching about 17% of the initial carbon at a CI of 4.9. This demonstrates the significant generation of volatile species during the UV exposure period. The volatile carbon release is likely to be linked to the generation of molecular-sized gaseous products such as alkane and alkene, peroxides, ketones, alcohols, acids, CO and $CO_2$ [19,40–43].

In addition, in Fig 3, the polymer mass loss reaches a plateau around ~10% of the initial plastic mass whereas carbon loss (corresponding to volatile compounds) increases regularly with UV aging.

Therefore, stabilization of the polymer mass loss does not indicate that the degradation process has stopped: it is a consequence of the partial compensation of carbon loss via the integration of oxygen atoms within the material. The significant discrepancy between carbon mass loss and polymer mass loss once again highlights the need to monitor degradation in terms of carbon balance, rather than relying on mass monitoring alone.

## Proportion of initial carbon retrieved in microplastics

The mass of carbon in MP is determined by measurement of the mass of generated MP collected in the syringe filter $m_{MPfilter,t}$ and by measurement of their average carbon mass fraction $\%C_{MP,t}$. Even considering the great care with which we performed the protocol, a small proportion of MP can stick to the glassware during the filtration step. These microplastics are taken into account by weighing the glassware before and after filtration and counting the difference as residual MP of which the mass $m_{MPres,t}$ is added to the mass $m_{MPfilter,t}$ measured on the filter, so that:

$$mC_{MP,t} = (m_{MPfilter,t} + m_{MPres,t}) \times \%C_{MP,t} \tag{6}$$

The results for MP collected in filters and residual MP on glassware are presented in Fig 4.

The threshold for MP release seems to be between CI of 2.5 and 4. After that threshold, the MP quantity increases rapidly, representing more than 55% of the initial Carbon mass content at a CI of 4.9.

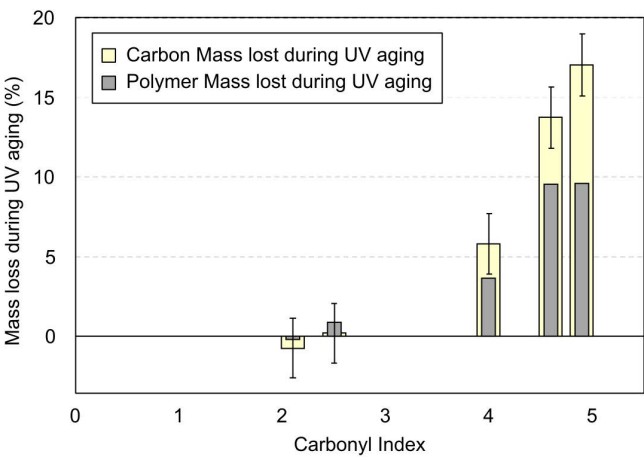

**Fig 3. Evolution of the proportion of carbon mass released as volatile compounds (in clear yellow) during weathering as a function of post-aging carbonyl index.** For comparison, the total polymer mass loss during weathering is represented (in dark grey).

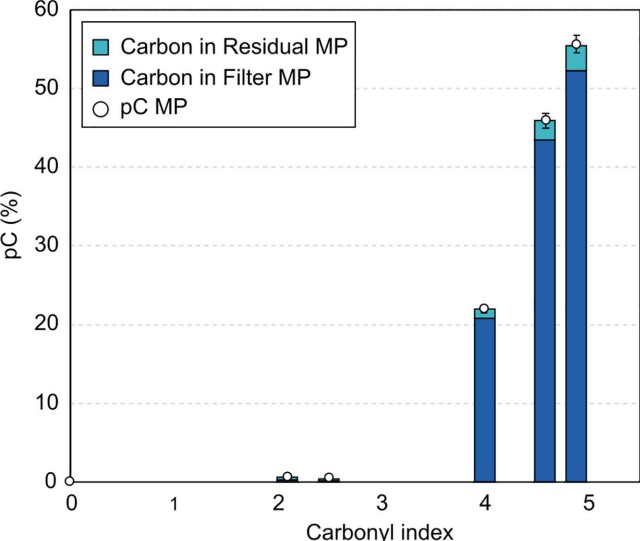

**Fig 4. Evolution of the proportion of initial Carbon retrieved in Microplastics as a function of post-aging carbonyl index.**

### Proportion of initial carbon retrieved in nanoplastics and soluble compounds

The carbon mass released by the granules in the form of NP or soluble molecules quantified from the filtrate (after filtration using a pore size of 1µm) is expressed by Eq. 7 and presented in Fig 5.

$$mC_{NP/Sol.,t} = V_{Filtrate,t} \times \gamma_{NP/Solubles,t}$$ (7)

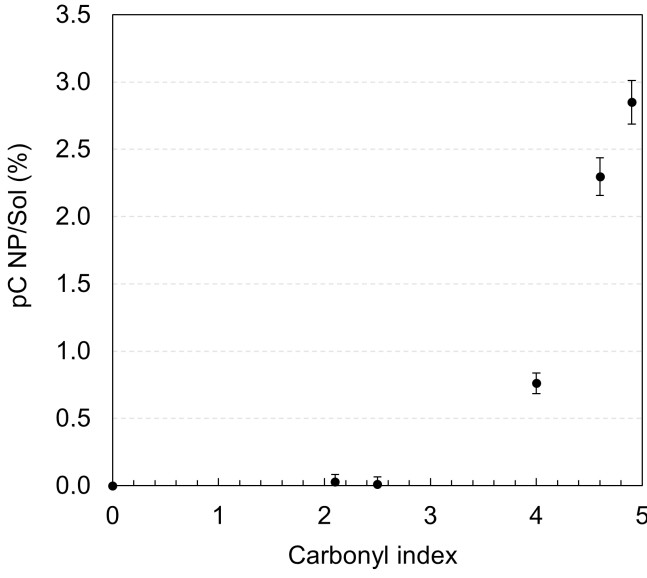

**Fig 5. Evolution of the proportion of initial Carbon retrieved in NP/Sol as a function of post-aging carbonyl index.**

No significant NP/Sol could be detected up to a CI of 2.5. For a higher oxidation level, the ratio of carbon retrieved in NP and soluble compounds ($pC_{NP/Sol}$), increases rapidly following the same trend as with regard to volatile compounds and MP, but in a much smaller proportion, reaching only 2.8% of the initial carbon at the higher CI studied here.

### Proportion of initial carbon remaining in the granules

In order to complete the carbon mass balance, it is necessary to measure the ratio of initial carbon remaining in the degraded granules ($pC_{GUV\omega,t}$). The values are displayed in Fig 6 as a function of CI.

About 100% of the carbon remains in the granules until the CI reaches 2.5, which is consistent with the observation that no substantial quantity of volatiles, MP, NP or soluble molecules is generated prior to this level of oxidation. This indicates that even if their surface is oxidized, the granules keep their structural integrity up to this CI. Above that, the carbon remaining in the granules decreases rapidly with only 21% remaining at CI 4.9. These results show that a minimum amount of photo-oxidation is required for PP to undergo any observable degradation, in alignment with the requirement for an oxidized surface layer sufficiently thick for MP and NP to be released.

## Discussion

All the results from the previous sections were combined to create a comprehensive carbon mass balance and to describe what happens to carbon during PP photo-degradation (Fig 7). A significant outcome of this study is the ability to track all carbon throughout the aging process, as we reach 100% of the initial carbon by adding up all the degradation products considered, notwithstanding their considerable range in size. This validates the newly developed carbon-based balance protocol and the hypothesis that the volatile compounds are essentially emitted during UV exposure.

The data on Fig 7 demonstrates that after a period during which no degradation product is measured (up to CI = 2.5) volatile, soluble and solid products (NP, MP) are released concomitantly in the atmosphere and surrounding aquatic system (from CI = 4).

One can interpret these results in the framework of the previously proposed scenarios [44–48]. An increased carbonyl index is indicative of the formation of an oxidation layer, which increases in thickness with exposure time, becoming

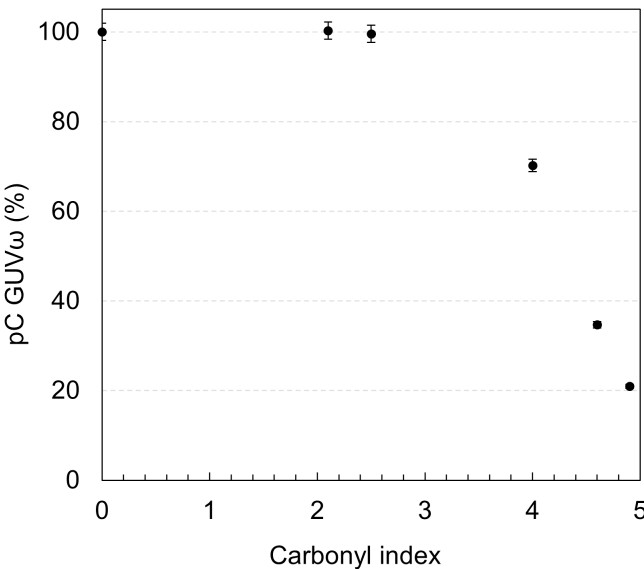

**Fig 6. Evolution of the proportion of initial Carbon retrieved in GUVω as a function of post-aging carbonyl index.**

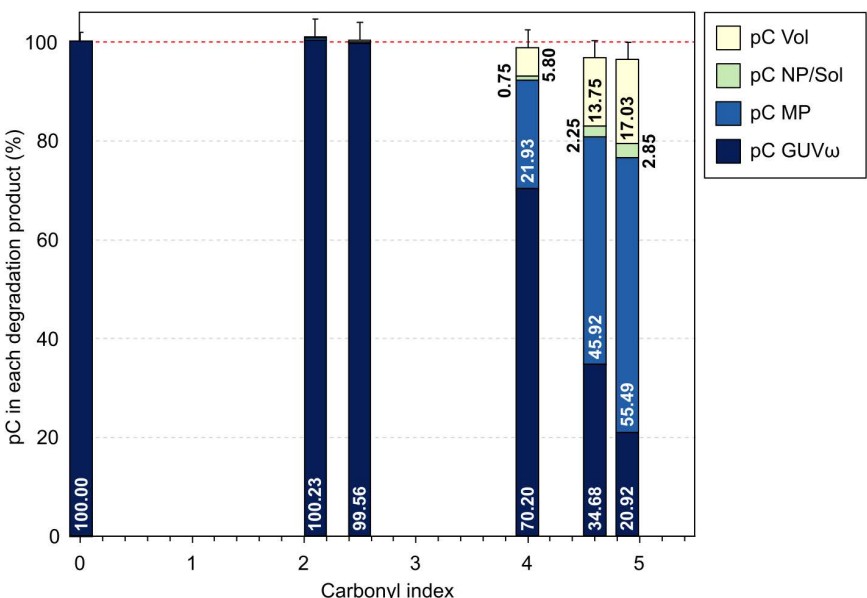

**Fig 7. Evolution of the complete quantitative carbon balance of PP abiotic degradation as a function of post-aging carbonyl index.**

increasingly brittle due to chemical modification and macromolecular cleavage. This, in turn, induces the creation of defects, recrystallization, and potential cross-linking. Additionally, macromolecular cleavage results in the release of small volatile or soluble molecules in quantifiable amounts over time. As the oxidized layer becomes increasingly brittle, whenever mechanical stresses are applied (here, stirring) it fragments into solid particles (MP and NP). While the mechanisms for obtaining small molecules and solid particles are distinct, both are contingent upon macromolecular cleavage. This may explain why the characteristic generation times of these products are of a similar order of magnitude. A further distinctive feature of the methodology employed in this study is that the quantity of volatile compounds produced during the aging process is expressed as a proportion of the carbon initially present in the polymer. Previous studies have shown that a significant amount of volatile or soluble molecules can be released during aging of various polymers [46,18,49], and the current data substantiates the assertion that they constitute a significant proportion of the degradation products (more than 15%, for the most oxidized samples). This underscores the importance of considering the release of volatile molecules, often wrongly neglected, in any assessment of plastic debris degradation products. Moreover, applying UV exposure and mechanical stirring one after the other, allows not only to quantify the amount of carbon released in the form of volatile molecules, but also to isolate the specific effect of stirring on the fragmentation of the surface layer and production of solid particles (MP and NP) [50]. The MP and NP generated are quite certainly dependent on the stirring strength and duration as well as on the thickness of the oxidation layer. The protocol used here allows us to analyze separately the respective effects of UV exposure and mechanical stirring. These parameters will be further tested in the future. Fig 7 demonstrates that at elevated degradation levels, the overall carbon balance becomes less accurate. The discrepancy in retrieved carbon can be attributed to two potential sources. Firstly, the production of volatile compounds is measured only during UV exposure. However, it is likely that during the stirring in water (whose acidity increases with degradation level [25]) the polymer releases gaseous compounds and is therefore not assessed during the COT experiment. A second reason could be that the amount of MP remaining on the filtration glassware increases with the total amount of released MP. However, based on meticulous measurements of the residual carbon fraction on the glassware, we think that the first phenomenon is more probable.

In order to avoid the issue of inhomogeneous UV exposure in the aging chamber, we have chosen to present the results in terms of the evolution of carbonyl index rather than exposure time. Even though, as explained by Rouillon et al. [51], CI is not a quantitative probe to monitor PP degradation, it still constitutes one of the most accurate indicators of the UV dose received by the sample. However, as there is no simple correlation between accelerated aging under UV exposure and real outdoor exposure, CI cannot be quantitatively related to time in the environment. It is important to note that even when examining the same polymer, there are inherent challenges in comparing degradation studies. Firstly, the manners in which UV exposure is achieved and, even more so, the methods by which external stresses are applied vary considerably depending on the article (in particular because pristine plastics of different shapes and sizes are used) [50,52]. Secondly, polymer degradation is assessed using a wide range of different methods. For example, Meides et al. [52] studied polypropylene degradation via reduction, after UV exposure and stirring in water, in the average size of a micrometric particle powder. It is not possible to directly correlate these results with an increased quantity of degradation products, as measured in the present study. While the methodology proposed here does not perfectly simulate the environment, it is nevertheless representative of the main mechanisms coming into play. As a result, it is a valuable tool for predicting how a given plastic will degrade in a specific environment, depending on its nature or characteristics. In particular, the protocol described here enables identification of the amount of carbon content released for each size class, which is an important piece of information, especially insofar as it is well-established that the impacts of plastic degradation are largely dependent on particle sizes [53]. Moreover, by comparing the kinetics of all the generated degradation products with their respective quantities, it could help to predict and compare the long-term degradation behavior of different plastics exposed to the same environmental conditions, and ultimately yield a safe-by-design approach.

In the case of PP, following an induction phase, the same sample will yield a significantly higher concentration of degradation products under prolonged exposure to UV radiation and subsequent abrasion. These findings reiterate the critical importance of plastic waste collection prior to its entry into aquatic environments, particularly when subjected to intense solar radiation. As demonstrated in this study, the prolonged presence of such PP granules on beaches may result in the formation of MP, NP, volatile compounds, and soluble molecules in the event of their entry into the aquatic system.

This study has demonstrated the effectiveness of a novel and complete method for quantitative assessment of polymer degradation products by monitoring carbon redistribution under volatile, soluble and particulate compounds. In a real-life environment, many other factors may influence the degradation process, such as biofouling, biodegradation or adsorption of dissolved organic matter. Yet, to the best of our knowledge, this is the first application of a carbon mass balance to predict what becomes of plastics undergoing abiotic degradation in a simplified artificial environment. Given the substantial chemical alterations inherent to oxidative processes, this methodology is considerably more robust and pertinent than the mere monitoring of degradation products. This methodology is applicable to all types of carbon-based polymers, whether or not they contain additives. Moreover, it can lead to more comprehensive understanding of the degradation mechanisms and to evaluation of the impact of each tested parameter. The abiotic degradation of plastics represents a complex environmental process, encompassing a multitude of stressors that are inherently challenging to replicate in laboratory settings. In the case of non-biodegradable plastics, photo-oxidation coupled with mechanical abrasion are understood to be the primary factors driving their degradation in the environment. Here, the weathering protocol was optimized to combine these two factors. Nevertheless, additional modifications to the type, parameters, and stirring medium may be made to enhance the environmental relevance of the protocol (e.g., the use of saltwater, a wave simulator, etc.). It is likely that the intensity of UV exposure and/or mechanical abrasion, as designed in this study, would result in modifications to the carbon mass balances calculated. Nevertheless, employing this protocol to compare plastic samples of different shapes and chemical nature will provide important insights into their degradation mechanisms and help to estimate and compare their potential impacts on the environment.

## Supporting information

**S1 SupInfo.** Informations on hydrogen content, lamp spectra, TOC calibration, mass corrections.
(DOCX)

**S2 Data.** Data for the realization of the figures including the carbon balance.
(XLSX)

## Author contributions

**Conceptualization:** Fabienne Lagarde.

**Data curation:** Fabien Boucher, Justine Gérome.

**Formal analysis:** Gustave Bertier, Arnaud Martel, Pascale Fabre, Fabien Boucher.

**Funding acquisition:** Matthieu George, Fabienne Lagarde.

**Investigation:** Matthieu George, Gustave Bertier, Pascale Fabre, Justine Gérome, Fabienne Lagarde.

**Methodology:** Matthieu George, Gustave Bertier, Arnaud Martel, Pascale Fabre, Fabien Boucher, Justine Gérome, Fabienne Lagarde.

**Supervision:** Matthieu George, Arnaud Martel, Pascale Fabre, Fabienne Lagarde.

**Validation:** Arnaud Martel, Pascale Fabre, Fabienne Lagarde.

**Writing – original draft:** Matthieu George, Gustave Bertier, Arnaud Martel, Pascale Fabre, Fabienne Lagarde.

**Writing – review & editing:** Matthieu George, Gustave Bertier, Arnaud Martel, Pascale Fabre, Fabienne Lagarde.

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
