## [Decision Letter · Decision Letter 0]

10 Jul 2025

PONE-D-25-25994Where does the carbon go? A new carbon balance method to assess the fate of plastics under solar exposurePLOS ONE?

Dear Dr. George,

Thank you for submitting your manuscript to PLOS ONE. After careful consideration, we feel that it has merit but does not fully meet PLOS ONE’s publication criteria as it currently stands. Therefore, we invite you to submit a revised version of the manuscript that addresses the points raised during the review process.

We look forward to receiving your revised manuscript.

Kind regards,

Wei-Chun Chin

Academic Editor

PLOS ONE

**Journal Requirements:**

1. When submitting your revision, we need you to address these additional requirements. Please ensure that your manuscript meets PLOS ONE's style requirements, including those for file naming. The PLOS ONE style templates can be found at https://journals.plos.org/plosone/s/file?id=wjVg/PLOSOne_formatting_sample_main_body.pdf and https://journals.plos.org/plosone/s/file?id=ba62/PLOSOne_formatting_sample_title_authors_affiliations.pdf 2. We suggest you thoroughly copyedit your manuscript for language usage, spelling, and grammar. If you do not know anyone who can help you do this, you may wish to consider employing a professional scientific editing service.  The American Journal Experts (AJE) (https://www.aje.com/) is one such service that has extensive experience helping authors meet PLOS guidelines and can provide language editing, translation, manuscript formatting, and figure formatting to ensure your manuscript meets our submission guidelines. Please note that having the manuscript copyedited by AJE or any other editing services does not guarantee selection for peer review or acceptance for publication.  Upon resubmission, please provide the following: The name of the colleague or the details of the professional service that edited your manuscript A copy of your manuscript showing your changes by either highlighting them or using track changes (uploaded as a *supporting information* file) A clean copy of the edited manuscript (uploaded as the new *manuscript* file) 3. Thank you for stating the following financial disclosure: PNR-EST CIPROP funded by ANSES.   Please state what role the funders took in the study.  If the funders had no role, please state: "The funders had no role in study design, data collection and analysis, decision to publish, or preparation of the manuscript." If this statement is not correct you must amend it as needed. Please include this amended Role of Funder statement in your cover letter; we will change the online submission form on your behalf. 4. Thank you for stating the following in the Acknowledgments Section of your manuscript: This work was supported by the PNR-EST Grant “CIPROP” funded by ANSES. We note that you have provided funding information that is not currently declared in your Funding Statement. However, funding information should not appear in the Acknowledgments section or other areas of your manuscript. We will only publish funding information present in the Funding Statement section of the online submission form. Please remove any funding-related text from the manuscript and let us know how you would like to update your Funding Statement. Currently, your Funding Statement reads as follows: PNR-EST CIPROP funded by ANSES.  Please include your amended statements within your cover letter; we will change the online submission form on your behalf. 5. Please amend either the abstract on the online submission form (via Edit Submission) or the abstract in the manuscript so that they are identical.

**Additional Editor Comments:**

Please revise the manuscript and reformat the manuscript based on the suggestions from the reviewers.

Reviewers' comments:

Reviewer's Responses to Questions

**Comments to the Author**

1. Is the manuscript technically sound, and do the data support the conclusions?

Reviewer #1: No

Reviewer #2: Yes

Reviewer #3: Yes

2. Has the statistical analysis been performed appropriately and rigorously?

Reviewer #1: N/A

Reviewer #2: N/A

Reviewer #3: Yes

3. Have the authors made all data underlying the findings in their manuscript fully available?

Reviewer #1: Yes

Reviewer #2: Yes

Reviewer #3: Yes

4. Is the manuscript presented in an intelligible fashion and written in standard English?

Reviewer #1: No

Reviewer #2: No

Reviewer #3: Yes

**Reviewer #1:**  I find the topic of this paper interesting. Before conducting a detailed review, I first took a general look to understand its contents. However, I believe that this paper is not yet ready for academic peer review. Chapter 2 is titled Methodology and Results, and Chapter 3 is Discussion, while Chapter 4 is Materials and Methods. It is strange to describe the methodology again in Chapter 4 when it has already been presented in Chapter 2. Even if the contents of the two chapters differ, this structure is highly unusual. Moreover, I have never seen a paper that presents both the methods and the results together in a single chapter like Chapter 2 does. I am quite puzzled by the authors’ reasoning behind this structure, and I find it hard to understand.

In addition, the methodology is described in a very superficial way. While editing is not the most critical factor in determining the quality of a paper, it should at least be edited sufficiently for reviewers to understand the content properly. However, there are many typographical and formatting errors. In fact, there is even a strange phrase on line 327: “Error! Reference source not found.” This suggests that the authors submitted the paper without even properly proofreading it. Regardless of the content of the paper, I believe it is not ready to undergo peer review in its current state. A substantial and thorough revision is necessary before resubmission.

**Reviewer #2:**  The manuscript describes a novel methodology to track the degradation of plastics upon aging, based on measurement of carbon release. The method will be very valuable for the scientific community and further research in the plastic field.

However, in general, the manuscript is difficult to read and understand in its current version, requiring a lot of going back and forth, to understand the steps that have been taken. Here I suggest to re-assess the current structure, with a method&result section coming first, followed by a detailed method section later on. For the latter, I also advise to describe the steps of the methods in the order they are conducted. In principle, I suggest to go for a individual results section.

Language needs refinement, some examples given below.

Some specific recommendations:

Line 26: delete "current"

Line 31: meaning of "and charges"?

Line 76: delete "in the last 4 years"

Line 101 (and more later on): link to reference

L133: "inherent heterogen. of UV chamber" this needs explaination, it is not fully addressed in the discussion. Why is there hetreogenicity?

Line 164: mention already here that open lids were required for oxygen supply.

line 250: delete "the quantity"

L325: specify whether the magnetic stirrer was covered with plastic, and what would be the consequences

L344: Was there a weight loss during cryo milling? If so, is it relevant for subsequent measurements?

Figure 3: implies that more carbon mass is lost than polymer mass. So, were there other sources of carbon?

Overall, I hence recommend a major revision of the manuscript.

**Reviewer #3:**  Although the problem of microplastics is currently addressed in hundreds or thousands of publications, this one can be published because it brings new information. The methodology of analyzing the C balance during degradation seems to be innovative and may be useful to other researchers. I do not have any critical remarks. I suggest the authors to re-edit the manuscript (see line 101)

**Do you want your identity to be public for this peer review?** For information about this choice, including consent withdrawal, please see our Privacy Policy

Reviewer #1: No

Reviewer #2: No

Reviewer #3: No

---

## [Author Response · Author response to Decision Letter 1]

26 Aug 2025

We answered to all comments rewritten here in the file "answer to reviewers."

Dear Editors and Reviewers,

Thank you for your useful comments and remarks. They are addressed in the revised form of the manuscript; the modifications made are detailed thereafter. We acknowledge that we did not explain clearly enough the innovative methodology and the first results that were obtained thanks to it. We have thus made a major revision of the paper, taking into account your recommendations. Moreover, we hope we corrected all the typos that appeared in the previous version.

>Reviewer #1: I find the topic of this paper interesting. Before conducting a detailed review, I first took a general look to understand its >contents. However, I believe that this paper is not yet ready for academic peer review.

We hope that this revised version qualifies now, since we improved the English and hopefully the clarity of the paper, by modifying its structure in particular.

>Chapter 2 is titled Methodology and Results, and Chapter 3 is Discussion, while Chapter 4 is Materials and Methods. It is strange to >describe the methodology again in Chapter 4 when it has already been presented in Chapter 2. Even if the contents of the two chapters >differ, this structure is highly unusual. Moreover, I have never seen a paper that presents both the methods and the results together in >a single chapter like Chapter 2 does. I am quite puzzled by the authors’ reasoning behind this structure, and I find it hard to >understand. In addition, the methodology is described in a very superficial way.

Structure was a major drawback of the previous version as were pointed out also by reviewer #2. Our original choice was guided by the necessity to present concomitantly the innovative experimental procedure and the original way to analyse the result by means of a carbon balance, as well as the first results obtained on polypropylene granules.

We have changed the structure towards a clearer and more usual one :

- A first section entitled “Experimental Section” which includes a sub-section describing the protocole as well as the details of the different techniques

- A second section entitled “Results and analysis” that explains how we track step by step where the carbon initially present in the polymer goes and which illustrates this methodology with the results obtained on aged polypropylene granules.

- A final section entitled “Discussion”

>While editing is not the most critical factor in determining the quality of a paper, it should at least be edited sufficiently for reviewers to >understand the content properly. However, there are many typographical and formatting errors. In fact, there is even a strange phrase >on line 327: “Error! Reference source not found.” This suggests that the authors submitted the paper without even properly >proofreading it. Regardless of the content of the paper, I believe it is not ready to undergo peer review in its current state. A substantial >and thorough revision is necessary before resubmission.

We apologize for the topographical and formatting errors that occurred during the submission process. The article has been carefully and thoroughly corrected. We also hired a professional English speaker to proofread and correct the article.

>Reviewer #2: The manuscript describes a novel methodology to track the degradation of plastics upon aging, based on measurement of >carbon release. The method will be very valuable for the scientific community and further research in the plastic field.

>However, in general, the manuscript is difficult to read and understand in its current version, requiring a lot of going back and forth, to >understand the steps that have been taken. Here I suggest to re-assess the current structure, with a method&result section coming >first, followed by a detailed method section later on. For the latter, I also advise to describe the steps of the methods in the order they >are conducted. In principle, I suggest to go for a individual results section.

Structure was a major drawback of the previous version as was pointed out also by reviewer #1. Our original choice was guided by the necessity to present concomitantly the innovative experimental procedure and the original way to analyse the result by mean of a carbon balance, as well as the first results obtained on polypropylene granules.

We have changed the structure towards a clearer and more usual one:

- A first section entitled “Experimental Section” which includes a sub-section describing the protocole as well as the details of the different techniques

- A second section entitled “Results and analysis” that explains how we track step by step where the carbon initially present in the polymer goes and which illustrates this methodology with the results obtained on aged polypropylene granules.

- A finale section entitled “Discussion”

Language needs refinement, some examples given below.

We also hired a professional English speaker to proofread and correct the article.

>Some specific recommendations:

>Line 26: delete "current"

Current has been deleted (Line 26)

>Line 31: meaning of "and charges"?

Charges refers to the organic or inorganic fillers that can sometimes be added to manufacture a plastic material. This could be misleading as the plastics we studied do not contain charges. We therefore removed “and charges” (Line 31)

>Line 76: delete "in the last 4 years"

In the last 4 years has been deleted (Line 77)

>Line 101 (and more later on): link to reference

All links to references have been checked and corrected

>L133: "inherent heterogeneity of UV chamber" this needs explanation, it is not fully addressed in the discussion. Why is there heterogeneity?

There is inherent heterogeneity in the ageing chamber as the artificial lights do not illuminate the samples with the same UV intensity depending of their position. In the range of possible positions, the samples more towards the middle tend to be more illuminated, resulting in potentially greater photo-oxidation or spontaneous inflammation and therefore greater abiotic degradation for a similar exposure time.

This point is stated in both “accelerated ageing chamber” sub-section (lines 152-158) and in “discussion” section (lines 358-362). Expressing the level of photo-oxidation in terms of the carbonyl index (CI) reached makes it possible to overcome this experimental limitation.

>Line 164: mention already here that open lids were required for oxygen supply.

This point is now stated clearly in the “overall description of the protocol” section (lines 119-121) which comes first in the revised structure.

>line 250: delete "the quantity"

“the quantity” has been replaced by “the amount” (line 345)

>L325: specify whether the magnetic stirrer was covered with plastic, and what would be the consequences

The magnetic stirrers used are indeed covered in PTFE. Each magnetic stirrer was thoroughly washed before use and no sign of degradation, or any wear was observed after use. Nevertheless, to avoid counting potential contamination during the NP assessment, the TOC results obtained on blanks that underwent the same protocol (values quickly becoming negligible compared to the quantities of NPs detected as soon as CI 4 is reached) were subtracted.

This has been added in the article (line 160-166)

>L344: Was there a weight loss during cryo milling? If so, is it relevant for subsequent measurements?

It must be noted, that cryo-milling is used exclusively to grind collected aged granules into powder, in order to perform elemental analysis and get averaged carbon content of these degraded products. To make that clearer, cryo-milling technical description has been incorporated into the “elemental analysis” section (lines 183-186).

Cryo-milling is thus done independently from the weight measurements. The only possible impact on the analysis would be a potential bias on the average carbon content, if the fraction lost on the walls of the cryo-miller was different from the rest. This fraction being very small (as we recovered almost all the sample), there is no reason to think that this is an actual issue. On the contrary, the cryo-milling step contributes to the agitation and homogenization of the sample in order to reduce as much as possible the bias on the average carbon content.

>Figure 3: implies that more carbon mass is lost than polymer mass. So, were there other sources of carbon?

This question concerns one of the main arguments underlying the protocol we have established. We therefore hope that things are clearer after these explanations and the rewriting of the article.

Figure 3 shows the mass proportion of carbon lost during photo-oxidation in the form of volatiles. For comparison, we have also shown in this figure the proportion of total mass lost by the plastic.

During photo-oxidation, two phenomena have opposite effects on the mass of aged plastic. On the one hand, some of the carbon initially contained in the plastic is lost in the form of volatile compounds, which reduces the mass. On the other hand, oxygen is incorporated into the polymer, which increases its mass.

The difference observed in Figure 3 between the two curves illustrates this and shows that looking only at the variation in polymer mass masks part of the phenomenon.

In other words, the difference between the two curves is solely due to the fact that the total mass loss of the polymer linked to carbon leakage is offset by a mass gain due to its oxidation. All the carbon in the degradation products comes from the polymer.

>Reviewer #3: Although the problem of microplastics is currently addressed in hundreds or thousands of publications, this one can be published because it brings new information. The methodology of analyzing the C balance during degradation seems to be innovative and may be useful to other researchers. I do not have any critical remarks. I suggest the authors to re-edit the manuscript (see line 101)

We apologize for the topographical and formatting errors that occurred during the submission process. The article has been carefully and thoroughly corrected. We also hired a professional English speaker to proofread and correct the article.

---

## [Decision Letter · Decision Letter 1]

9 Sep 2025

Where does the carbon go? A new carbon balance method to assess what happens to plastics under solar exposure

PONE-D-25-25994R1

Dear Dr. George,

We’re pleased to inform you that your manuscript has been judged scientifically suitable for publication and will be formally accepted for publication once it meets all outstanding technical requirements.

Kind regards,

Wei-Chun Chin

Academic Editor

PLOS ONE

Additional Editor Comments (optional):

Reviewer #1:

Reviewer #3:

Reviewers' comments:

Reviewer's Responses to Questions

**Comments to the Author**

Reviewer #1: All comments have been addressed

Reviewer #3: All comments have been addressed

2. Is the manuscript technically sound, and do the data support the conclusions?

Reviewer #1: Yes

Reviewer #3: Yes

3. Has the statistical analysis been performed appropriately and rigorously?

Reviewer #1: Yes

Reviewer #3: Yes

4. Have the authors made all data underlying the findings in their manuscript fully available?

Reviewer #1: Yes

Reviewer #3: Yes

5. Is the manuscript presented in an intelligible fashion and written in standard English?

Reviewer #1: Yes

Reviewer #3: Yes

Reviewer #1: I think the revision is good. Now I recommend to accept this manuscript. Thanks for the authors' efforts.

Reviewer #3: The work adds interesting information to the current state of knowledge. The authors have responded to my comments. The work may be published.

**Do you want your identity to be public for this peer review?** For information about this choice, including consent withdrawal, please see our Privacy Policy

Reviewer #1: No

Reviewer #3: No

---

## [Editor Report · Acceptance letter]

PONE-D-25-25994R1

PLOS ONE

Dear Dr. George,

I'm pleased to inform you that your manuscript has been deemed suitable for publication in PLOS ONE. Congratulations! Your manuscript is now being handed over to our production team.

Kind regards,

on behalf of

Dr. Wei-Chun Chin

Academic Editor

PLOS ONE